# LOCAL BINARY PATTERN NETWORKS FOR CHARACTER RECOGNITION

## ABSTRACT

Memory and computation efficient deep learning architectures are crucial to the continued proliferation of machine learning capabilities to new platforms and systems, especially, mobile sensing devices with ultra-small resource footprints. In this paper, we demonstrate such an advance for the well-studied character recognition problem. We use a strategy different from the existing literature by proposing local binary pattern networks or LBPNet that can learn and perform bit-wise operations in an end-to-end fashion. Binarization of operations in convolutional neural networks has shown promising results in reducing the model size and computing efficiency. Characters consist of some particularly structured strokes that are suitable for binary operations. LBPNet uses local binary comparisons and random projection in place of conventional convolution (or approximation of convolution) operations, providing important means to improve memory and speed efficiency that is particularly suited for small footprint devices and hardware accelerators. These operations can be implemented efficiently on different platforms including direct hardware implementation. LBPNet demonstrates its particular advantage on the character classification task where the content is composed of strokes. We applied LBPNet to benchmark datasets like MNIST, SVHN, DHCD, ICDAR, and Chars74K and observed encouraging results.

## INTRODUCTION

Convolutional Neural Networks (CNN) (LeCun et al., 1989a) have had a notable impact on many applications. Modern CNN architectures such as AlexNet (Krizhevsky et al., 2012), VGG (Simonyan & Zisserman, 2015), GoogLetNet (Szegedy et al., 2015), and ResNet (He et al., 2016) have greatly advanced the use of deep learning techniques (Hinton et al., 2006) into a wide range of computer vision applications (Girshick et al., 2014; Long et al., 2015). As deep learning models mature and take on increasingly complex pattern recognition tasks, these demand tremendous computational resources with correspondingly higher performance machines and accelerators that continue to be fielded by system designers. It also limits their use to applications that can afford the energy and/or cost of such systems. By contrast, the universe of embedded devices especially when used as intelligent edge-devices in the emerging distributed systems presents a higher range of potential applications from augmented reality systems to smart city systems.

Optical character recognition (OCR) particularly in the wild, shown in Fig. 1, has become an essential task for computer vision applications such as autonomous driving and mixed reality. There existed CNN-based methods (Yin et al., 2013) and other probabilistic learning methods (Yao et al., 2014a;b) handling the OCR tasks. However, the CNN-based models are computation demanding, and the probabilistic learning methods required more patches, e.g., empirical rule, clustering, error correction, or boosting to improve accuracy.

Various methods have been proposed to perform network pruning (LeCun et al., 1989b; Guo et al., 2016), compression (Han et al., 2015; Iandola et al., 2016), or sparsification(Liu et al., 2015). Impressive results have been achieved lately by using binarization of selected operations in CNNs (Courbariaux et al., 2015; Hubara et al., 2016; Rastegari et al., 2016). At the core, these efforts seek to approximate the internal computations from floating point to binary while keeping the underlying convolution operation exact or approximate, but the nature of character images has not been fully utilized yet.

We propose LBPNet as a light-weighted and compact deep-learning approach that can leverage the nature of character images since LBPNet is sensitive to discriminative outlines and strokes. Precisely, we focus on the task of character classification by exploring an alternative using non-convolutional operations that can be executed in an architectural and hardware-friendly manner, trained in an end-to-end fashion from scratch (distinct to the previous attempts of binarizing the CNN operations). We note that this work has roots in research before the current generation of deep learning methods. Namely, the adoption of local binary patterns (LBP) (Ojala et al., 1996), which uses a number of *predefined* sampling points that are mostly on the perimeter of a circle, to compare with the pixel value at the center. The combination of multiple logic outputs ("1" if the value on a sampling point is greater than that on the center point and "0" otherwise) gives rise to a surprisingly rich representation (Wang et al., 2009) about the underlying image patterns and has shown to be complementary to the SIFT-kind features (Lowe, 2004). However, LBP has been under-explored in the deep learning research community where the feature learning part in the existing deep learning models (Krizhevsky et al., 2012; He et al., 2016) primarily refers to the CNN features in a hierarchy. We found LBP operations particularly suitable in recognizing characters that consist of structured strokes. Despite recent attempts such as (Juefei-Xu et al., 2017), the logic operation (comparison) in LBP has not been used in the existing CNN frameworks due to the intrinsic difference between the convolution and comparison operations.

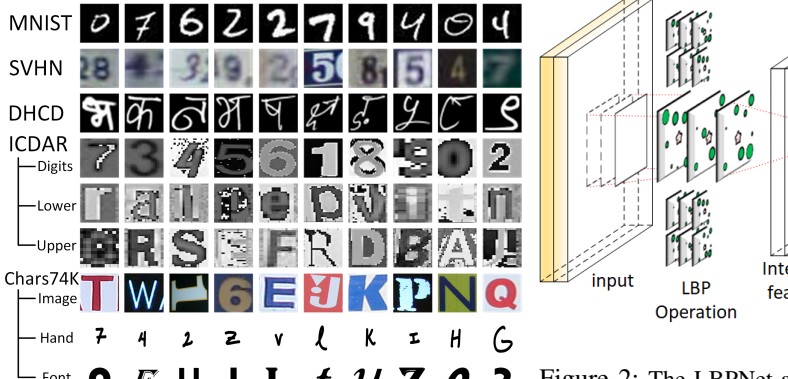

Figure 1: Examples in character recognition datasets.

Figure 2: The LBPNet architecture. The LBP operation generates feature maps with comparison and bit-allocation, while random projection fuses the intermediate channels.

Several features make LBPNet distinct from previous attempts. All the binary logic operations in LPBNet are directly learned, which is in a stark distinction to previous attempts that try to either binarize CNN operations (Hubara et al., 2016; Rastegari et al., 2016) or to approximate LBP with convolution operations (Juefei-Xu et al., 2017). Further, the LBP kernels in previous works are fixed upon initialized because the lack of a suitable mechanism to train the sampling patterns. Instead, we derive a differentiable function to learn the binary pattern and adopt random projection for the fusion operations. Fig. 2 illustrates the overview of LBPNet. The resulting LBPNet is very suitable for the character recognition tasks because the comparison operation can capture and comprehend the sharp outlines and distinct strokes among character images.Experiments show that thus configured LBP-Net achieves the state-of-the-art results on benchmark datasets while accomplishing a significant improvement in the parameter size reduction gain (hundreds) and speedup (thousand times faster). That means LBPNet efficiently utilizes every storage bit and computation unit through the learning of image representations.

## RELATED WORKS

Related works regarding model reduction of CNN fall along four primary dimensions.

**Character recognition**. Besides CNN-based methods for character recognition like BNN (Hubara et al., 2016), random forest (Yao et al., 2014a;b) was prevailing as well. However, the random forest methods usually required one or more techniques such as feature extraction, clustering, or error correction codes to improve the recognition accuracy. Our method, instead, provides a compact end-to-end and computation efficient solution to character recognition.

**Binarization for CNN**. Binarizing CNNs to reduce the model size has been an active research direction (Courbariaux et al., 2015; Hubara et al., 2016; Rastegari et al., 2016). Through binarizing both weights and activations, the model size was reduced, and a logic operation can replace the multiplication. Non-binary operations like batch normalization with scaling and shifting are still in floating-point (Hubara et al., 2016). The XNOR-Net (Rastegari et al., 2016) introduces extra scaling layer to compensate for the loss of binarization and achieves a state-of-the-art accuracy on ImageNet. Both BNNs and XNORs can be considered as the discretization of real-numbered CNNs, while the core of the two works is still based on spatial convolution.

**CNN approximation for LBP operation**. Recent work on local binary convolutional neural networks (LBCNN) in (Juefei-Xu et al., 2017) takes an opposite direction to BNN (Hubara et al., 2016). LBCNN utilizes subtraction between pixel values together with a ReLU layer to simulate the LBP operations. During the training, the sparse binarized difference filters are fixed, only the successive 1-by-1 convolution, serving as channel fusion mechanism and the parameters in batch normalization layers, are learned. However, the feature maps of LBCNN are still in floating-point numbers, resulting in significantly increased model complexity as shown in Table 2. By contrast, LBPNet learns binary patterns and logic operations from scratch, resulting in orders of magnitude reduction in the memory size and an increase in testing speed over LBCNN.

**Active or deformable convolution**. Among the notable line of recent work that learns local patterns are active convolution (Jeon & Kim, 2017) and deformable convolution (Dai et al., 2017), where data dependent convolution kernels are learned. Both of these are quite different from LBPNet since they do not seek to improve network efficiency. Our binary patterns learn the position of the sampling points in an end-to-end fashion as logic operations (without the need for the use of addition operations). By contrast, directly relevant earlier work (Dai et al., 2017) essentially learns data-dependent convolutions.

## LOCAL BINARY PATTERN NETWORK

Fig. 2 shows an overview of the LBPNet architecture. The forward propagation is composed of two steps: LBP operation and channel fusion. We introduce the patterns in LBPNets and the two steps in the following sub-sections and then describe the engineered network structures for LBPNets.

### PATTERNS IN LBPNETS

In LBPNet, multiple patterns defining the positions of sampling points generate multiple output channels. Patterns are randomly initialized with a uniform distribution of locations centered on a predefined square window, and then subsequently learned in an end-to-end supervised learning fashion. Fig. 3 (a) shows a traditional local binary pattern, which is a fixed

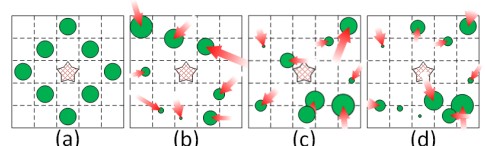

Figure 3: (a) A traditional local binary pattern. (b)-(d) Our learnable local binary patterns. The red arrows denote pushing forces during training.

pattern without much variety; there are eight sampling points denoted by green circles, surrounding a pivot point in the meshed star at the center of pattern; Fig. 3(b)-(d) shows a learnable pattern with eight sampling points in green and a pivot point as a star at the center. Our learnable patterns are initialized using a normal distribution of positions within a given area. Different sizes of the green circle stand for the bit position of the comparison outcome on the output bit array. We allocate the comparison outcome of the largest green circle to the most significant bit of the output pixel, the second largest to the second largest bit, and so on. The red arrows represent the driving forces that can push the sampling points to better positions to minimize the classification error. The model size of an LBPNet is tiny compared with CNN because the learnable parameters in LBPNets are the sparse and discrete sampling patterns.

### LBP OPERATION

First, LBPNet samples pixels from incoming images and compares the sampled pixel value with the center sampled point, the pivot. If the sampled pixel value is larger than that of the center one, the output is a bit "1"; otherwise, the output is set to "0." Next, we allocate the output bits to a binary digit array in the output pixel based on a predefined ordering. The number of sampling points defines the number of bits of an output pixel on a feature map. Then we slide the local binary pattern to the

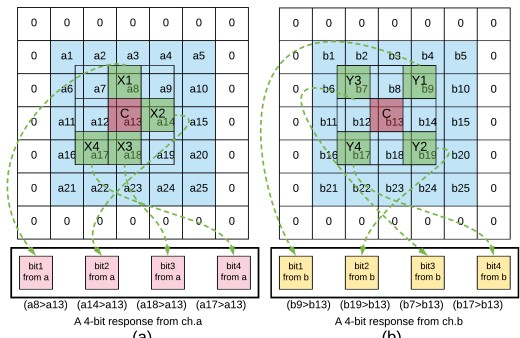 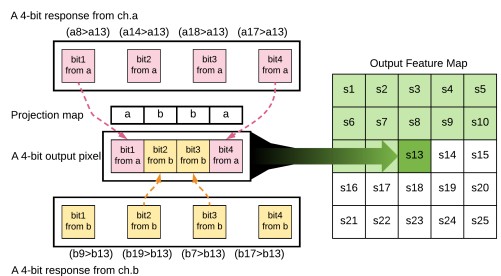

Figure 4: An example of an LBP operation on multiple input channels. LBP operations for channel (a) ch.a and (b) ch.b. Each pattern has four sampling points restricted in a 3-by-3 area.

Figure 5: An example of LBP channel fusing. The two 4-bit responses from Fig. 3 are fused and assigned to pixel s13 on the output feature map.

next location and perform the aforementioned steps until a feature map is generated. In most cases, the incoming image has multiple channels; hence we perform the LBP operation on every input channel.

Fig. 4 shows a snapshot of the LBP operations. Given two input channels, ch.a and ch.b, we perform the LBP operation on each channel with different kernel patterns. The two 4-bit response binary numbers of the intermediate output are shown on the bottom. For clarity, we use green dashed arrows to mark where the pixels are sampled and list the comparison equations under the resulting bits. A logical problem has emerged: we need a channel fusion mechanism to avoid the explosion of the exponential growing channel numbers.

CHANNEL FUSION WITH RANDOM PROJECTION

We use random projection (Bingham & Mannila, 2001) as a dimension-reducing and distance-preserving process to select output bits among intermediate channels for the concerned output channel as shown in Fig. 5. The random projection is implemented with a predefined mapping table for each output channel, i.e., we fix the projection map upon initialization. All output pixels on the same output channel share the same mapping. Random projection

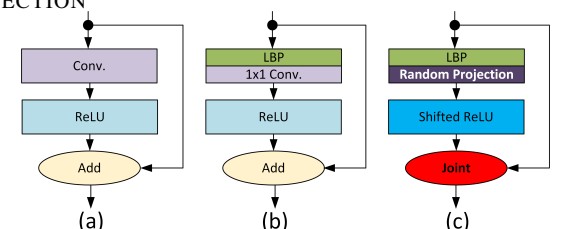

Figure 6: Basic LBPNet blocks. (a) the well-known building block of residual networks. (b) The transition-type building block uses a 1-by-1 convolutional layer for the channel fusion of a preceding LBP layer. (c) The multiplication and accumulation (MAC) free building block for LBPNet.

not only solves the channel fusion with a bit-wise operation but also simplifies the computation, because we do not have to compare all sampling points with the pivots. For example, in Fig. 5, the two pink arrows from intermediate ch.a, and the two yellow arrows from intermediate ch.b bring the four bits for the composition of an output pixel. Only the MSB and LSB on ch.a and the middle two bits on the ch.b need to be computed. If the output pixel is $n$-bit, for each output pixel, there will be $n$ comparisons needed, which is irrelevant to the number of input channels. The more input channels bring the more combinations of representations in a random projection table.

Throughout the forward propagation, there are no multiplication or addition operations. Only comparison and memory access are used. Therefore, the design of LBPNets is efficient in the aspects of both software and hardware.

NETWORK STRUCTURES FOR LBPNET

The network structure of LBPNet must be carefully designed. Owing to the nature of the comparison, the outcome of an LBP layer is very similar to the outlines in the input image. In other words, our LBP layer is good at extracting high-frequency components in the spatial domain but relatively weak at understanding low-frequency components. Therefore, we use a residual-like structure to compensate for this weakness of LBPNet. Fig. 6 shows three kinds of residual-net-like building

blocks. Fig. 6 (a) is the typical building block for residual networks. The convolutional kernels learn to obtain the residual of the output after the addition. Our first attempt is to introduce the LBP layer into this structure as shown in Fig. 6 (b), in which we utilize a 1-by-1 convolution to learn a combination of LBP feature maps. However, the convolution incurs too many multiplication and accumulation operations especially when the LBP kernels increases. Then, we combine LBP operation with a random projection as shown in Fig. 6 (c). Because the pixels in the LBP output feature maps are always positive, we use a shifted rectified linear layer (shifted-ReLU) to increase nonlinearities. The shifted-ReLU truncates any magnitudes below half of the maximum of the LBP output. More specifically, if a pattern has $n$ sampling points, the shifted-ReLU is defined as Eq. 1.

$$f(x) = \begin{cases} x & , x > 2^{n-1} - 1 \\ 2^{n-1} - 1 & , \text{otherwise} \end{cases} \tag{1}$$

As mentioned earlier, the low-frequency components reduce as the information passes through several LBP layers. To preserve the low-frequency components while making the block MAC-free, we introduce a joint operation cascading the input tensor of the block and the output tensor of the shifted-ReLU along the channel dimension. The number of channels is under controlled since the increasing trend is linear to the number of input channels.

HARDWARE BENEFITS

LBPNet saves in hardware cost by avoiding the convolution operations. Table 1 lists the reference numbers of logic gates of the concerned arithmetic units. A ripple-carry full-adder requires 5 gates for each bit. A 32-bit multiplier includes a data-path logic and a control logic. Because there are too many feasible implementations of the control logic circuits, we

Table 1: The number of logic gates for arithmetic units. Energy use data for technology node: 45nm.

| Device | #bits | #gates | Energy (J) | #cycle |
|---|---|---|---|---|
| Adder | 4 | 20 | $\leq$ 3E-14 | 1 |
| | 32 | 160 | 9E-13 | 1 |
| Multiplier | 32 | $\geq$144 | 3.7E-12 | 4 |
| Comparator | 4 | 11 | $\leq$ 3E-14 | 1 |

conservatively use an open range to express the sense of the hardware expense. The comparison can be made with a pure combinational logic circuit of 11 gates, which also means only the infinitesimal internal gate delays dominate the computation latency. The comparison is not only cheap regarding its gate count but also fast due to a lack of sequential logic inside. Slight difference in numbers of logic gates may apply if different synthesis tools or manufacturers are chosen. With the capability of an LBP layer as strong as a convolutional layer concerning classification accuracy, replacing the convolution operations with comparison gives us a 27X saving of hardware cost.

Another important benefit is energy saving. The energy demand for each arithmetic device has been shown in (Horowitz, 2014). If we replace all convolution operations with comparisons, the energy consumption is reduced by 153X.

Moreover, the core of LBPNet is composed of bit shifting and bitwise-OR, and both of them have no concurrent accessing issue. If we are implementing an LBPNet hardware accelerator, no matter on FPGA or ASIC flow, the absence of the concurrent issue resulted from convolution's accumulation process will guarantee a speedup over CNN hardware accelerator. For more justification, please refer to the forward algorithm in the appendix.

BACKWARD PROPAGATION OF LBPNET

To train LBPNets with gradient-based optimization methods, we need to tackle two problems: 1). The non-differentiability of comparison; and 2). The lack of a source force to push the sampling points in a pattern.

DIFFERENTIABILITY

The first problem can be solved if we approximate the comparison operation with a shifted and scaled hyperbolic tangent function as shown in Eq. 2.

$$I_{lbp} > I_{pivot} \stackrel{approximated}{\rightarrow} \frac{1}{2}(tanh(\frac{I_{lbp} - I_{pivot}}{k}) + 1), \tag{2}$$

where $k$ is the scaling parameters to accommodate the number of sampling points from a previous LBP layer, $I_{lbp}$ is the sampled pixel in a learnable LBP kernel, and $I_{pivot}$ is the sampled pixel on the pivot. We provide a sensitivity analysis of $k$ w.r.t. classification accuracy in the appendix. The hyperbolic tangent function is differentiable and has a simple closed-form for the implementation.

DEFORMATION WITH OPTICAL FLOW THEORY

To deform the local binary patterns, we resort to the concept from optical flow theory. Assuming the image content in the same class share the same features, even though there are certain minor shape transformations, chrominance variations or different view angles, the optical flow on these images should share similarities with each other. $\frac{\partial I}{\partial x}V_x + \frac{\partial I}{\partial y}V_y = -\frac{\partial I}{\partial t}$ The equation above shows the optical flow theory, where $I$ is the pixel value, a.k.a luminance, $V_x$ and $V_y$ represent the two orthogonal components of the optical flow among the same or similar image content. The LHS of optical flow theory can be interpreted as a dot-product of image gradient $(\frac{\partial I}{\partial x}\hat{x} + \frac{\partial I}{\partial y}\hat{y})$ and optical flow $(V_x\hat{x} + V_y\hat{y})$, and this product is the negative derivative of luminance versus time across different images, where $\hat{x}$ and $\hat{y}$ denote the two orthogonal unit vectors on the 2-D coordinate.

To minimize the difference between images in the same class is equivalent to extract similar features of the images in the same class for classification. However, both the direction and magnitude of the optical flow underlying the dataset are unknown. The minimization of a dot-product cannot be done by changing the image gradient to be orthogonal with the optical flow. Therefore, the only feasible path to minimize the magnitude of the RHS is to minimize the image gradient. Please note the sampled image gradient can be changed by deforming the apertures, which are the sampling points of local binary patterns.

When applying calculus chain rule on the cost of LBPNet with regard to the position of each sampling point, one can easily conclude that the last term of the chain rule is the image gradient. Since the sampled pixel value is the same as the pixel value on the image, the gradient of sampled value with regard to the sampling location on a pattern is equivalent to the image gradient on the incoming image. Eq. 3 shows the gradient from the output loss through a fully-connected layer with weights, $w_j$, toward the image gradient.

$$\frac{\partial cost}{\partial position} = \sum_j (\Delta_j w_j)\frac{\partial g(s)}{\partial s}\frac{\partial s}{\partial I_{lbp}}(\frac{\mathbf{d}I_{lbp}}{\mathbf{d}x}\hat{x} + \frac{\mathbf{d}I_{lbp}}{\mathbf{d}y}\hat{y}), \tag{3}$$

where $\Delta_j$ is the backward propagated error, $\frac{\partial g(s)}{\partial s}$ is the derivative of activation function, and $\frac{\partial s}{\partial I_{lbp}}$ is the gradient of Eq. 2. Please refer to the appendix for more details of the forward-backward training algorithm.

EXPERIMENTS

In this section, we conduct a series of experiments on five datasets and their subsets: MNIST, SVHN, DHCD, ICDAR2005, and Chars74K to verify the capability of LBPNet. Some typical images of these character datasets are shown in Fig. 1. Please refer to the appendix for the description of datasets. We additionally evaluate LBPNet on a few broader categories such as face, pedestrian, and affNIST and have observed promising results for object classification.

EXPERIMENT SETUP

In all of the experiments, we use all training examples to train LBPNets and directly validate on test sets. To avoid peeping, we do not employ the validation errors in the backward propagation. There are no data augmentations used in the experiments.

We implement two versions of LBPNet using the two building blocks shown in Fig. 6 (b) and (c). For the remaining parts of this paper, we call the LBPNet using 1-by-1 convolution as the channel fusion mechanism **LBPNet(1x1)** (has convolution in the fusion part), and the version of LBPNet utilizing random projection **LBPNet(RP)** (totally convolution-free). The number of sampling points in a pattern is set to 4, and the area size for the pattern to deform is 5-by-5.

LBPNet also has an additional multilayer perceptron (MLP) block, which is made with two fully-connected layers of 512 and #classes neurons. Besides the nonlinearities, there is one batch-normalization layer. The MLP block's performance without any convolutional layers or LBP layers on the three datasets is shown in Table 2, 3. The model size and speed of the MLP block are excluded in the comparisons since all models have an MLP block.

To understand the capability of LBPNet when compared with existing convolution-based methods, we build two feed-forward streamline CNNs as our baseline for each dataset. **CNN-baseline** is designed in the same number of layers and number of kernels with the LBPNet; the other, **CNN-lite**, is

designed subject to the same memory footprint with the LBPNet(RP). The basic block of the CNNs contains a spatial convolution layer (Conv) followed by a batch normalization layer (BatchNorm) and a rectified linear layer (ReLU).

In the BNN (Hubara et al., 2016) paper, the classification on MNIST is done with a binarized multilayer perceptron network (MLP). We adopt the binarized convolutional neural network (BCNN) in (Hubara et al., 2016) for SVHN to perform the classification and re-produce the same accuracy as shown in (Lin et al., 2017) on MNIST.

EXPERIMENTAL RESULTS

Table 2 and 3 show the experimental results of LBPNet on MNIST and SVHN together with the baseline and previous works. We list the classification error rate, model size, latency of the inference, and the speedup compared with the baseline CNN. The best value of each column is shown in bold. Please note the calculation of latency in cycles is made with an assumption that no SIMD parallelism and pipelining optimization is applied. Because we need to understand the total number of computations in every network but both floating-point and binary arithmetics are involved, we cannot use FLOPs as a measure. Therefore, we adopt typical cycle counts shown in Table 1 as the measure of latencies. For the calculation of model size, we exclude the MLP blocks and count the required memory for necessary variables to focus on the comparison between the intrinsic operations in CNNs and LBPNets, respectively the convolution and the LBP operation.

Table 2: The performance of LBPNet on MNIST.

| | Error ↓ | Size ↓ (Bytes) | Latency ↓ (cycles) | Speedup ↑ |
|---|---|---|---|---|
| MLP Block | 24.22% | - | - | - |
| CNN-baseline | **0.44%** | 1.41M | 222.0M | 1X |
| CNN-lite | 1.20% | 456 | **553K** | **401.4X** |
| BCNN | 0.47% | 1.89M | 306.1M | 0.725X |
| LBCNN | 0.49% | 12.2M | 8.78G | 0.0253X |
| LBPNet (this work) | | | | |
| LBPNet (1x1) | 0.50% | 1.27M | 27.73M | 8.004X |
| LBPNet (RP) | 0.50% | **397.5** | 651.2K | 340.8X |

Table 3: The performance of LBPNet on SVHN.

| | Error ↓ | Size ↓ (Bytes) | Latency ↓ (cycles) | Speedup ↑ |
|---|---|---|---|---|
| MLP Block | 77.78% | - | - | - |
| CNN-baseline | 8.30% | 15.96M | 9.714G | 1X |
| CNN-lite | 69.14% | 2.80K | **1.576M** | **6164X** |
| BCNN | **2.53%** | 1.89M | 312M | 31.18X |
| LBCNN | 5.50% | 6.70M | 7.098G | 1.369X |
| LBPNet (this work) | | | | |
| LBPNet (1x1) | 8.33% | 1.51M | 9.175M | 1059X |
| LBPNet (RP) | 7.31% | **2.79K** | 4.575M | 2123X |

**MNIST**. The CNN-baseline and LBPNet(RP) share the same network structure, 39-40-80, and the CNN-lite is limited to the same memory size so that the network structure is 2-3. The baseline CNN achieves the lowest classification error rate 0.44%. The BCNN possesses a decent speedup while maintaining the classification accuracy. While LBCNN claimed its saving in memory footprint, to achieve 0.49% error rate, 75 layers of LBCNN basic blocks are used. As a result, LBCNN loses speedups. The 3-layer LBPNet(1x1) with 40 LBP kernels and 40 1-by-1 convolutional kernels achieves 0.50%. The 3-layer LBPNet(RP) reaches 0.50% error rate as well. Although LBPNet's performance is slightly inferior, the model size of LBPNet(RP) is reduced to 397.5 bytes, and the speedup is 340.8X faster than the baseline CNN. Even BCNN cannot be on par with such a vast memory reduction and speedup. The CNN-lite delivering the worst error rate demonstrates that if we shrink a CNN model down to the same memory size as the LBPNet(RP), the classification error of CNN(lite) is greatly sacrificed.

**SVHN**. Table 3 shows the experimental results of LBPNet on SVHN together with the baseline and previous works. The CNN-baseline and LBPNet(RP) share the same network structure, 67-70-140-280-560, and the CNN-lite is limited to the same memory size so that the network structure is 8-17. BCNN outperforms our baseline and achieves 2.53% with smaller memory footprint and higher speed. LBCNN also achieve a good memory reduction and 1.369X speed-up. The 5-layer LBPNet(1x1) with 8 LBP kernels and 32 1-by-1 convolutional kernels achieve 8.33%, which is close to our baseline CNN's 8.30%. The convolution-free LBPNet(RP) for SVHN is built with 5 layers of LBP basic blocks, 67-70-140-280-560, and achieves 7.31% error rate. Compared with CNN(lite)'s high error rate, the learning of LBPNet's sampling point positions is proven to be effective and economical.

**More Results.** Table 4 lists the experimental results of LBPNet(RP) on all character recognition datasets. LBPNets achieve the state-of-the-art accuracies on all of the datasets.

PRELIMINARY RESULTS ON OBJECTS AND DEFORMED PATTERNS
Next, we show results on datasets of general objects.

**Pedestrain**: We first evaluate LBPNet on the INRIA pedestrian dataset (Dalal & Triggs, 2005), which consists of cropped positive and negative images. Note that we did not im-

Table 4: LBPNet structures and experimental results.

|  | Type | Structure | Error↓ | Size↓ | Reduction ↑ | Latency ↓ | Speedup ↑ |
|---|---|---|---|---|---|---|---|
| MNIST | CNN-3L | 39-40-80 | 0.44% | 1.41M | - | 222.0M | - |
|  | LBPNet(RP) | 39-40-80 | 0.50% | 397.5 | 3547X | 651.2K | 341X |
| SVHN | CNN-5L | 67-70-140-280-560 | 8.30% | 15.96M | - | 9.714G | - |
|  | LBPNet(RP) | 67-70-140-280-560 | 7.31% | 2.79K | 5720X | 4.575M | 348X |
| DHCD | CNN | 63-64-128-256 | 0.72% | 4.61M | - | 1.59G | - |
|  | LBPNet(RP) | 63-64-128-256 | 0.81% | 1.28K | 3602X | 2.973M | 535X |
| ICDAR-Digits | CNN | 3-4 | 0.00% | 44.47K | - | 556.6K | - |
|  | LBPNet(RP) | 3-4 | 0.00% | 317 | 140X | 11.95K | 47X |
| ICDAR-LowerCase | CNN | 3-4 | 0.00% | 44.47K | - | 556.6K | - |
|  | LBPNet(RP) | 3-4 | 0.00% | 317 | 140X | 11.95K | 47X |
| ICDAR-UpperCase | CNN | 3-4 | 0.00% | 44.47K | - | 556.6K | - |
|  | LBPNet(RP) | 3-4 | 0.00% | 317 | 140X | 11.95K | 47X |
| Chars74K-EnglishImg | CNN | 63-64-128-256-512 | 40.54% | 12.17M | - | 6.218G | - |
|  | LBPNet(RP) | 63-64-128-256-512 | 41.69% | 2.56K | 4754X | 4.19M | 1,484X |
| Chars74K-EnglishHnd | CNN | 63-64-128 | 28.68% | 1.95M | - | 434.5M | - |
|  | LBPNet(RP) | 63-64-128 | 26.63% | 637.5 | 3059X | 1.044M | 416X |
| Chars74K-EnglishFnt | CNN | 63-64-128 | 21.91% | 1.95M | - | 434.5M | - |
|  | LBPNet(RP) | 63-64-128 | 22.74% | 637.5 | 3059X | 1.044M | 416X |

plement an image-based object detector due to the focus of our paper. Fig. 7 shows the trade-off curves of a 3-layer LBPNet (37-40-80) and a 3-layer CNN (37-40-80). Here we did not exhaustively explore the capability of LBPNet for object classification.

**Face**: We apply our LBPNet on FDDB dataset (Jain & Learned-Miller, 2010) to verify the face classification performance of LBPNet. Same as previously, we perform training and testing on a dataset of cropped images; we use the annotated positive face examples with cropped four non-person frames in every training image to create negative face examples for both training and testing. The structures of the LBPNet and CNN are the same as before (37-40-80). LBPNet achieves 97.78%, and the baseline CNN reaches 97.55%.

**affNIST**: We conduct an experiment on affNIST [1], which is composed of 32 translation variations of MNIST (including the original MNIST). To accelerate the experiment, we randomly draw three variations of each original example to get training and testing subsets of affNIST. We repeat the same process to draw examples and train

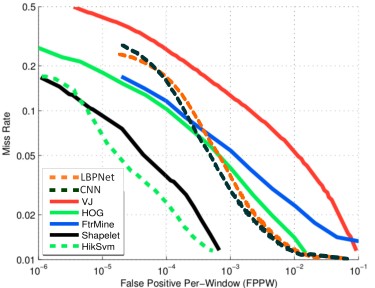

Figure 7: The classification error trade-off (DET) curves of a 3-layer LBPNet and a 3-layer CNN on the INRIA pedestrian dataset (Dalal & Triggs, 2005). We plot the results on Fig.8(a) of Dollár et al. (2009) for comparison with the other five approaches.

the networks ten times to get an averaged result. The network structure of LBPNet and our baseline CNN are the same, 39-40-80. To improve the translation invariant property of the networks, we use two max-pooling layers following the first and second LBP layer or convolutional layer. With the training and testing on the subsets of affNIST, LBPNet achieves 93.18%, and CNN achieves 94.88%.

## CONCLUSION AND FUTURE WORK

We have built a convolution-free, end-to-end, and bitwise LBPNet from basic operations and verified its effectiveness on character recognition datasets with orders of magnitude speedup (hundred times) in testing and model size reduction (thousand times) when compared with the baseline and the binarized CNNs. The learning of local binary patterns results in an unprecedentedly efficient model since, to the best of our knowledge, there is no compression/discretization of CNN can achieve the KByte level model size while maintaining the state-of-the-art accuracy on the character recognition tasks. Both the memory footprints and computation latencies of LBPNet and previous works are listed. LBPNet points to a promising direction for building new generation hardware-friendly deep learning algorithms to perform computation on the edge devices.

---

[1]https://www.cs.toronto.edu/ tijmen/affNIST/

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

APPENDIX

FORWARD PROPAGATION ALGORITHM

---

**Algorithm 1:** Forward of LBPNet

---

**input** : An input tensor $X$ of shape $(c_i, w, h)$, previous pattern $P$ of shape $(c_o, n_s)$, and the fixed projection map $M$ of shape $(c_o, n_s)$. The pattern width $k$ and padding width $d = \left\lfloor \frac{k}{2} \right\rfloor$. Please note every element of $P$ is a tuple.

**output**: A scalar predictions $y$.

---

1   $X \leftarrow \text{ZeroPadding}(X, d)$;
2   **for** $i_o = 1$ *to* $c_o$ **do**
3     **for** $i_h = $ *to* $h$ **do**
4       **for** $i_w = 1$ *to* $w$ **do**
5         **for** $i_s = 1$ *to* $n_s$ **do**
6           $i_i \leftarrow M[i_o, i_s]$;
7           $(i_{px}, i_{py}) \leftarrow P[i_o, i_s]$;
8           $\text{pivot} \leftarrow X[i_w + d][i_w + d][i_i]$;
9           $\text{sample} \leftarrow X[i_w + i_{px}][i_w + i_{py}][i_i]$;
10          **if** *sample* > *pivot* **then**
11            $y[i_w][i_h][i_o] \mathrel{|}= 1 \ll i_s$
12          **end**
13         **end**
14       **end**
15     **end**
16   **end**
17   return $y$

---

Alg. 1 describes the forward algorithm of an LBP layer. The three outermost nested loops form the sliding window operation to generate an output feature maps, and the innermost loop is the LBP operation. We combine the LBP operation with random projection to skip unnecessary comparisons. Firstly, we look up the random projection map for the input plane index and then use it to sample only the necessary pairs for the comparison.

The core of LBPNet is implemented with bit shifting and bitwise-OR, and both of them have no concurrent accessing issue. That is, we can directly implement it with CUDA programming to accelerate the inference on GPU. If we are implementing an LBPNet hardware accelerator, no matter on FPGA or ASIC flow, the absence of concurrent issue resulted from CNN's accumulation process will guarantee a speedup over CNN's hardware accelerator.

BACKWARD PROPAGATION ALGORITHM

---

**Algorithm 2:** Backward of LBPNet

---

**input** : An input tensor $X$, a gradient tensor of loss w.r.t the output of current layer $go_{(c_o \times w_o \times h_o)}$, previous pattern $P$, and the fixed projection map $M$. The pattern width $k$ and padding width $d$. During training, we remember the previous real-valued pattern $R$ of the same shape of $P$.

**output**: The gradient of loss w.r.t. the input tensor $g_i$ in shape $(c_i, w, h)$, and the gradient of loss w.r.t. the position of sampling point. $g_P$ in shape $(c_o, n_s)$. Please note every element of $g_P$ is a tuple.

---

1   $\bigtriangledown \leftarrow \text{ImageGradient}(X)$;
2   $P \leftarrow round(R)$;
3   $D \leftarrow \text{LookUpDifference}(X, P, M)$;
4   $E \leftarrow \text{ConstructExp}(\tanh(D), P, M)$;
5   $dE \leftarrow \text{ConstructDiffExp}(1\text{-}\tanh^2(D), P, M)$;
6   $g_i \leftarrow \frac{1}{2} g_o^T E$;
7   $g_P \leftarrow g_o(dE \bigstar \bigtriangledown)^T$;
8   return $g_i, g_P, R, P$

---

Alg. 2 describes the backward propagation at a high-level point of view. Because LBPNet requires sophisticated element-wise matrix operation, some of them have no matrix-to-vector or matrix-to-

matrix multiplication equivalence but can be implemented and optimized in low-level CUDA codes for training speed. The $ImageGradient(.)$ function calculates the image gradient vector field of the input feature map. Then, $round(.)$ function discretize the previous real-valued pattern for the image sampling later on. $LookUpDifference(.)$ samples the input tensor with the concerned input plane index from the projection map. This step is similar to the core of Alg. 1, but we calculate the difference instead of comparing the pairs of sampled pixels.

The $ConstructExp(.)$ function multiplies the hyperbolic tangential difference matrix with the exponential of 2 corresponding to the position of the comparison result in an output bit array. For example, if a comparison result is allocated to the MSB, the hyperbolic tangential value will be multiplied with $2^{n_s}$, assuming $n_s$ sampling pairs per kernel. The $ConstructDiffExp(.)$ performs the same calculation with $ConstructExp(.)$ except for the first argument is replaced with the derivative of $tanh(.)$. These two sub-routine functions convert sparse kernels to dense kernels for the follow matrix-to-matrix multiplications.

The sixth line uses a matrix-to-matrix multiplication to collect and weight the output gradient tensor from the successive layer. This step is the same with CNN's backward propagation. The resulting tensor is also called input gradient tensor and will be passed to the preceding layer to accomplish the backward propagation.

The seventh line element-wisely times the differential exponential matrix with the image gradient first and then multiply the result with the output gradient tensor. The resulting tensor carries the gradient of LBP parameters, $\frac{\partial cost}{\partial position}$, which will be multiplied with an adaptive learning rate for the update of sampling positions of an LBP kernel.

DATASET DESCRIPTIONS

Images in the padded MNIST dataset are hand-written numbers from 0 to 9 in 32-by-32 grayscale bitmap format. The dataset is composed of a training set of $60,000$ examples and a test set of $10,000$ examples. Both staff and students wrote the manuscripts. Most of the images can be easily recognized and classified, but there is still a portion of sloppy images inside MNIST.

SVHN is a photo dataset of house numbers. Although cropped, images in SVHN include some distracting numbers around the labeled number in the middle of the image. The distracting parts increase the difficulty of classifying the printed numbers. There are $73,257$ training examples and $26,032$ test examples in SVHN.

Table 5: The datasets we used in the experiment.

| | Description | #Class | #Examples | CNN Baseline | LBPNet (RP) (ours) |
|---|---|---|---|---|---|
| DHCD | Handwritten Devanagari characters | 46 | 46x2,000 | 98.47% (Acharya et al., 2015) | **99.19%** |
| ICDAR-DIGITS | Photos of numbers | 10 | 988 | 100.00% | 100.00% |
| ICDAR-UpperCase | Photos of lower case Eng. char. | 26 | 5,288 | 100.00% | 100.00% |
| ICDAR-LowerCase | Photos of upper case Eng. char. | 26 | 5,453 | 100.00% | 100.00% |
| Chars74K-EnglishImg | Photos, Alphanumeric | 62 | 7,705 | 47.09% (De Campos et al., 2009) | **58.31%** |
| Chars74K-EnglishHnd | Handwritten, Alphanumeric | 62 | 3,410 | 71.32% | **73.37%** |
| Chars74K-EnglishFnt | Printed Fonts, Alphanumeric | 62 | 62,992 | **78.09%** | 77.26% |

LEARNING CURVES

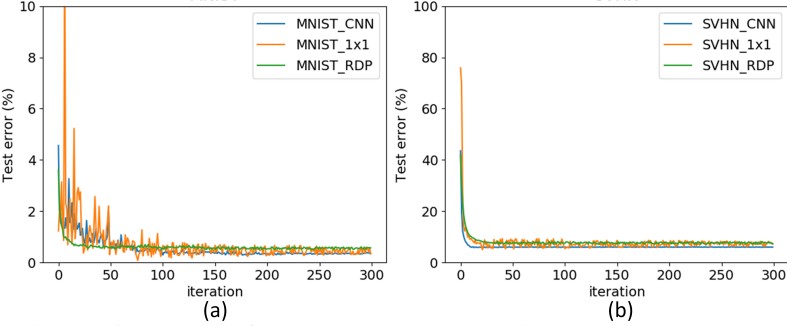

Fig. 8 shows the learning curves of LBPNets on MNIST and SVHN.

Figure 8: Error curves on benchmark datasets. (a) test errors on MNIST; (b) test errors on SVHN.

SENSITIVITY ANALYSIS OF $k$

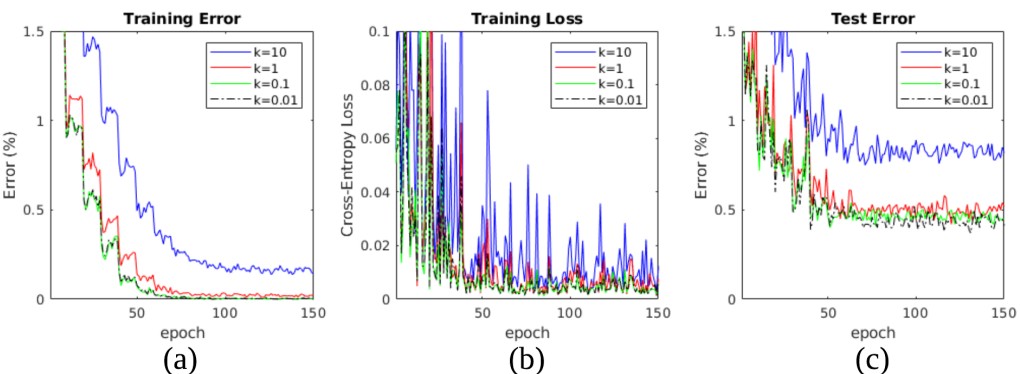

Figure 9: Sensitivity analysis of $k$ w.r.t. training error on MNIST. (a) Training error; (b) training loss; (c) test error.

Fig 9 shows the sensitivity analysis of the parameter $k$ in Eq. 2 w.r.t. the training accuracy. The LBPNet structure we use is 3-layer, 39-40-80. We gradually reduce $k$ from 10 to 0.01 to verify the effect on the learning curves. Sub-figure (a) and (c) shows the smaller $k$ is, the lower the error rate is, but there exist a saturation when $k$ decreases below 1. Sub-figure (b) shows a smaller $k$ suppresses the ripple of training loss better. As a summary, because we approximate the comparison function with a sifted and scaled hyperbolic tangent function. A smaller $k$ implies less error between the approximation and the original comparison curve, and hence simulate the comparison while securing differentiability. In this paper, we choose $k = 0.1$ to balance between classification accuracy and the overflow risk of the gradient summation during backward propagation.

