# OpenReview forum: "Local Binary Pattern Networks for Character Recognition"
_ICLR.cc/2019/Conference_

### Official Review · AnonReviewer3 · 2018-10-29
**interesting idea, but quite confused on implementation and not convinced on experiments**

**Rating:** 5
**Confidence:** 4

**Review:**

This paper proposed a LBPNet for character recognition, which introduces the LBP feature extraction into deep learning. Personally I think that this idea is interesting for improving the efficiency of CNNs, as traditionally LBP has been demonstrated its good performance and efficiency in some vision tasks such as face recognition or pedestrian detection. However, I do have the following concerns about the paper:

1. Calculation/Implementation of Eq. 4: I do not quite understand how it derived, and how to use Eq. 3 in calculation. I suggest the authors to explain more details, as this is the key for implementation of LBP layers.

2. Effects of several factors on performance in the experiments are missing: (1) random projection map in Fig. 5, (2) $k$ in Eq. 2, and (3) the order of images for computing RHS of Eq. 3. In order to better demonstrate LBPNet, I suggest to add such experiments, plus training/testing behavior comparison of different networks.

3. Does this network work with more much deeper?

4. Data: The datasets used in the experiments are all well-aligned. This makes me feel that the RHS of Eq. 3 does make sense, because it will capture the spatial difference among data, like temporal difference in videos. How will the network behave on the dataset that is not aligned well, like affnist dataset?

5. How will this network behave for the applications such as face recognition or pedestrian detection where traditionally LBP is applied?

---

> ### Author Response · Authors · 2018-11-15
> **We appreciate your valuable feedback. Please see our answers below.**
>
> [from authors:]
> Q: "Calculation/implementation of the chain rule equation."
> A: Please refer to the backward algorithm in the appendix. For more details regarding the engineering efforts, please refer to our reply to the second reviewer's question about the backpropagation.
>
> Q: "Effects of (1) random projection map (2) scaling factor k (3) the order of images for computing RHS of optical flow theory."
> A: (1) The random projection map is only initialized before the training of LBPNet. We draw samples from a uniform distribution of all channels. We have explored the Gaussian and uniform distribution and observed that the simple uniform distribution giving every channel the same chance delivered better classification accuracy. (2) Please refer to the sensitivity analysis of k in the appendix. (3) The optical flow theory only provides an intuition beneath the chain rule equation, and the RHS of optical flow theory is not calculated. Moreover, the stochastic gradient descent framework will shuffle the order of images to enforce the learning irrelevant to the order.
>
> Q: "Can LBPNet go deeper?"
> A: We have implemented an end-to-end LBPLayer module for stacking a deeper network but didn't observe much improvement with a few more layers. We can stack up to ten LBP layers by adding pooling layers to reduce the size of feature maps to fit in the VRAM. To make the networks even deeper, say 100 layers, more low-level coding will be involved. Convlayer is supported by the cuDNN library, and hence can be stacked up to a hundred layers. We are making efforts to make the LBPLayer module work with cuDNN so that the volatile VRAM can be used more efficiently.
>
> Q: "Performance on affNIST, face recognition and pedestrian detection."
> A: This is a good point which has also been raised by the other reviewers. Please see our reply to the previous reviewers and also see the newly-added section "Preliminary Results on Object and Deformed Patterns" on page 8.

---

### Official Review · AnonReviewer1 · 2018-11-03
**Empirically weak, practical advantage wrt to literature unclear.**

**Rating:** 6
**Confidence:** 4

**Review:**

In this work, a neural network that uses local binary patterns instead of kernel convolutions is introduced. Using binary patterns has two advantages: a) it reduces the network definition to a set of binary patterns (which requires much less storage than the floating point descriptions of the kernel weights used in CNNs) and b) allows for fast implementations relying only on logical operations (particularly fast on dedicated hardware).

This work is mostly descriptive of a proposed technique with no particular theoretical performance guarantees, so its value hinges mostly on its practical performance on real data. In that sense, its evaluation is relatively limited, since only figures for MNIST and SVHN are provided.

A list of additional datasets is provided in Table 5, but only the performance metric is listed, which is meaningless if it is not accompanied with figures for size, latency and speedup. The only takeway about the additional datasets is that the proposed LBPNet can match or outperform a weak CNN baseline, but we don't know if the latter achieves state-of-the-art performance (previous figures of the baseline CNN suggest it doesn't) and we don't know if there's significant gain in speed or size.

Regarding MNIST and SVHN, which are tested in some more detail, again, we are interested in the performance-speed (or size) tradeoff, and it is unclear that the current proposal is superior. The baseline CNN does not achieve state of the art performance (particularly in SVHN, for which the state-of-the-art is 1.7% and the baseline CNN achieves 6.8%). For SVHN, BCNN has a much better performance-speed tradeoff than the baseline, since it is both faster and higher performance. Then, the proposed method, LBPNet, has much higher speed, but lower performance than BCNN. It is unclear how LBPNet's and BCNN's speeds would compare if we were to match their performances. For this reason, it is unclear to me that LBPNet is superior to BCNN on SVHN.

Also the numbers in boldface are confusing, aren't they just incorrect for both the Latency and Error in MNIST? Same for the Latency in SVHN.

The description of the approach is reasonably clear and clarifying diagrams are provided. The backpropagation section seems a bit superficial and could be improved. For instance, backpropagation is computed wrt the binary sampling points, as if these were continuous, but they have been defined as discrete before. The appendix contains a bit more detail, where it seems that backpropagation is alternated with rounding. It's not justified why this is a valid gradient descent algorithm.

Also how the scaling k of the tanh is set is not explained clearly. Do you mean that with more sampling points k should be larger to keep the outputs of the approximate comparison operator close to 0 and 1?

Minor:

What exactly in this method makes it specific to character recognition? Since you are trying to capture both high-level and low-level frequencies, it seems you'd be capturing all the relevant information. SVHN data are color images with objects (digits) in it, what is the reason that makes other objects not be detectable with this approach?

English errors are pervasive throughout the paper. A non-exhaustive list:

Fig 4.b: X2 should be Y2
particuarly
"to a binary digits"
"In most case"
"0.5 possibility"
"please refer to Sec .."
"FORWARD PROPATATION"

---

> ### Author Response · Authors · 2018-11-15
> **We appreciate your valuable feedback. Please see our answers below.**
>
> [from authors:]
> Q: "A list of additional datasets is provided in Table 5, but only the performance metric is listed, which is meaningless if it is not accompanied by figures for size, latency, and speedup."
> A: We have updated figure 1 to indicate typical images from different datasets used in the paper. We have also changed Table 5 to a new Table 4 on page 8 with details about the network structures, error rates, memory sizes and latencies for these datasets. The baseline CNNs are set to be in the same structure as LBPNets to illustrate the benefit in terms of memory reduction and speedup.
>
> Q: "It is unclear to me that LBPNet is superior to BCNN on SVHN."
> A: Please note that BCNN and LBCNN rely on batch normalization after every binarized convolutional layer to boost accuracy, and the batch normalization layer is still real-numbered. The best quantization for the batch normalization reported in the literature is 16-bit to achieve loss-less performance, but our LBPNet and is built to be innately bitwise. On MNIST, DHCD, ICDAR, Chars74K and additionally evaluated pedestrian, face, and affNIST, the performance of LBPNet is comparable to CNN and BCNN. SVHN is the only dataset that has a slightly large gap. However, the memory size of LBPNet(RP) is 2.79KB, which is 677X smaller than the BCNN baseline.
>
> Q:" The numbers in boldface are confusing."
> A: The bold-face number is the best value in each column.
>
> Q: "The backpropagation section seems a bit superficial and could be improved. For instance, backpropagation is computed wrt the binary sampling points, as if these were continuous, but they have been defined as discrete before."
> A: The rounding process of LBPNet is similar with the weight binarization process in BNN, which applied the straight-through estimator using hard tanh function to replace the non-differentiable binarization. We also adopt the idea of the straight-through estimator to bypass and truncate the calculation of gradient over the rounding processing.
>
> In the appendix, we provide a high-level description of the backpropagation. The following paragraph describes a more detailed engineering work of the backpropagation. We follow the framework of ConvLayer's BLAS implementation, which leverages hardware supported im2col, col2im, GEMM, and GEMV primitive functions. Unlike ConvLayer having fixed kernels during gradient accumulation before the weight update so that the gradient w.r.t output (g_o) can be accumulated with a GEMM function, LBPLayer has sparse and irregular sampling patterns, which must be converted to equivalent dense kernels for GPU acceleration, and those kernels changes according to different location of the sliding window because of sampling. To remedy this changing kernel issue, we dive into low-level CUDA-C and directly calculate and allocate the intermediate tensors D, E and dE in an im2col format as described in the appendix. Then, we apply GEMM between g_o and the element-wise product of dE and image gradient tensor to calculate the gradient w.r.t. sampling position (g_P), and another GEMM between g_o and E to is to calculate the gradient w.r.t input (g_i) in an im2col format, and finally a col2im function call accumulates the local gradients covered by an kernel window into g_i.
>
> In brief, the calculation and implementation of the chain rule equation 3 involve linear algebra, approximation, and mostly GPU programming techniques since LBPNet is neither a CNN-based method nor realizable with any off-the-shelf libraries directly. We combine Torch7 (for common layers like ReLU and pooling) with BLAS library (for hardware supported primitives like GEMM, GEMV, im2col, and col2im) and low-level CUDA-C programming (for bit-wise comparison and pixel-wise allocation) to implement LBPNet.
>
> Q: "How the scaling k of the tanh is set is not explained clearly?"
> A: The scaling factor k is independent with the number of sampling points. We provide a sensitivity analysis of k with regard to the training accuracy in the appendix. In short, k is used to approximate the comparison operation while securing differentiability. Through the sensitivity analysis, we can balance between the approximation error and the risk of overflow during gradient accumulation.
>
> Q: "What exactly in this method makes it specific to character recognition? What is the reason that makes other objects not be detectable with this approach?"
> A: As we have mentioned in the introduction section, OCR tasks are well-suited for both LBP and LBPNet because the characters are made of strokes, edges, and stair-like patterns. The application of LBPNet is not limited to OCR, and we are reporting results on object recognition including pedestrians, face, and deformed patters. Please refer to the new section "Preliminary Results on Object and Deformed Patterns" on page 8.
>
> Q: "English errors."
> A: Thanks for pointing them out. We have revised them accordingly in the paper.

---

### Official Review · AnonReviewer2 · 2018-11-04
**Borderline paper**

**Rating:** 5
**Confidence:** 5

**Review:**

1. The paper introduces the idea of some existing hand-crafted features into the deep learning framework, which is a smart way for building light weighted convolutional neural networks.

2. I have noticed that binary patterns used in the paper are trainable, which means that these binary patterns can be seen as learned convolution filters with extremely space and computational complexity. Thus, the proposed method can also be recognized as a kind of binary network.

3.  The baseline BCNN has a different architecture to the network using the proposed method. Thus, comparisons shown in Table 3 and Table 4 are somewhat unfair.

4. The capability of the proposed method was only verified on character recognition datasets. Does it can be easily used for other tasks such as face recognition or object detection on some relatively large datasets?

---

> ### Author Response · Authors · 2018-11-15
> **We appreciate your valuable feedback. Please see our answers below.**
>
> [from authors:]
> Q:  "The baseline BCNN has a different architecture to the network using the proposed method. Thus, comparisons shown in Table 3 and Table 4 are somewhat unfair."
> A:  The BCNN results were copied from the existing papers and reported here. We keep the BCNN network unmodified and only calculate the corresponding memory footprint and latency for a comparison. Please note that BCNN and LBCNN rely on batch normalization after every binarized convolutional layer to boost accuracy, and the batch normalization layer is still real-numbered. The best quantization for the batch normalization reported in the literature is 16-bit to achieve loss-less performance, but our LBPNet and is built to be innately bitwise. Therefore, we mainly compare with the CNN-baseline to demonstrate the benefit of LBPNet regarding memory reduction and speedups. On MNIST, the classification error of BCNN and LBPNet are similar, yet the BCNN's memory size is 4755X larger than that of LBPNet. On SVHN, there exists an accuracy gap between BCNN and LBPNet, but the memory size of LBPNet(RP) is 2.79KB, which is 677X smaller than the BCNN baseline. We are continuing to improve the result on SVHN. The experimental results on the other datasets including newly experimented objects further show that the performance of LBPNet being on par with CNN.
>
> Q:  "Does it can be easily used for other tasks such as face recognition or object detection on some relatively large datasets?"
> A: This is a good point. To answer this question, we evaluated LBPNet on three additional datasets including the INRIA pedestrian dataset, the FDDB face dataset, and the affNIST dataset, and have observed encouraging results. We report them in a new section "Preliminary Results on Objects and Deformed Patterns" on page 8. LBPNet indeed achieves results on par with CNN but at a significantly reduced computation and model complexity, similar to the character case. This validates the effectiveness of LBPNet on broader object types beyond characters.

---

### Meta-Review · Area_Chair1 · 2018-12-17
**unconvincing**

**Confidence:** 4
**Recommendation:** Reject

**Metareview:**

This paper proposed a LBPNet for character recognition, which introduces the LBP feature extraction into deep learning. Reviewers are confused on implementation and not convinced on experiments. The only score 6 reviewer is also concerned "Empirically weak, practical advantage wrt to literature unclear". Only evaluating on MNIST/SVHN etc is not convincing to demo the effectiveness of the proposed method.